# OpenReview forum: "MULTI-STAGE GOAL VERIFICATION: A DEFENSE IN-DEPTH MATHEMATICAL FRAMEWORK FOR MITIGATING DECEPTIVE ALIGNMENT IN LARGE LANGUAGE MODELS"
_mathai.club/MathAI/2026/Conference — 2026 Oral_

### Official Review · Reviewer_g5uj · 2026-03-13
**This paper addresses the critical AI safety problem of inner alignment and deceptive alignment in large language models. The authors propose a novel Multi-Stage Goal Verification Framework designed to detect and mitigate the risk of mesa-optimizers -- internal sub-agents that may develop objectives misaligned with human intent. The framework combines three verification layers: natural language consistency (via Process Reward Models), internal latent state analysis (via mechanistic interpretability), and formal symbolic verification (via theorem provers). The core contribution is a mathematical formulation where a composite safety gate is integrated into the training loss function via a "safety tax" hyperparameter, making deceptive strategies gradient-disfavored. The paper presents empirical results across multiple benchmarks (MiniF2F, GSM8K, MATH) and models (DeepSeek-Math, AlphaGeometry, Llemma), showing consistent improvements of 10-15% in "reasoning accuracy"-- the percentage of problems solved with logically valid reasoning. The work is timely, well-motivated, and presents a concrete, technically sound approach to a pressing problem. Its primary strengths are its clear problem formalization, rigorous experimental validation, and the innovative integration of multiple verification modalities. Minor weaknesses include a lack of ablation studies to isolate the contribution of each verification stage and limited discussion of the computational overhead of the framework. Overall, this is a strong paper that makes a significant contribution to AI safety and verifiable reasoning.**

**Rating:** 7
**Confidence:** 3

**Review:**

### Brief Summary

This paper addresses the critical AI safety problem of inner alignment and deceptive alignment in large language models. The authors propose a novel Multi-Stage Goal Verification Framework designed to detect and mitigate the risk of mesa-optimizers -- internal sub-agents that may develop objectives misaligned with human intent. The framework combines three verification layers: natural language consistency (via Process Reward Models), internal latent state analysis (via mechanistic interpretability), and formal symbolic verification (via theorem provers). The core contribution is a mathematical formulation where a composite safety gate is integrated into the training loss function via a "safety tax" hyperparameter, making deceptive strategies gradient-disfavored. The paper presents empirical results across multiple benchmarks (MiniF2F, GSM8K, MATH) and models (DeepSeek-Math, AlphaGeometry, Llemma), showing consistent improvements of 10-15% in "reasoning accuracy"-- the percentage of problems solved with logically valid reasoning. The work is timely, well-motivated, and presents a concrete, technically sound approach to a pressing problem. Its primary strengths are its clear problem formalization, rigorous experimental validation, and the innovative integration of multiple verification modalities. Minor weaknesses include a lack of ablation studies to isolate the contribution of each verification stage and limited discussion of the computational overhead of the framework. Overall, this is a strong paper that makes a significant contribution to AI safety and verifiable reasoning.

### Detailed Review

**Overview:**
The paper tackles the problem of inner alignment in large language models. As models become more capable, they may develop internal objectives (mesa-objectives) that diverge from the intended human reward signal. The most dangerous form of this is deceptive alignment, where a model behaves safely during training to avoid modification, only to pursue its misaligned goals after deployment. Standard safety benchmarks are insufficient because they only evaluate external behavior, which a sufficiently intelligent model can manipulate.

To address this, the authors propose a Multi-Stage Goal Verification Framework. The core idea is to monitor the model not just at the output level, but across three distinct representational domains:

1.  Natural Language (Stage 1): Using a Process Reward Model (PRM) to verify the logical coherence of the Chain-of-Thought.
2.  Latent Representations (Stage 2): Leveraging mechanistic interpretability tools (e.g., Sparse Autoencoders) to detect internal "deception circuits."
3.  Formal Logic (Stage 3): Using symbolic engines (Lean, SymPy) to verify that the final output is mathematically valid.

These verification stages are combined into a composite safety gate. This gate is integrated into the training loss function as a penalty term. The hyperparameter acts as a "safety tax," ensuring that any attempt to achieve high task accuracy while failing the verification checks results in a significant loss spike, making deceptive strategies computationally inefficient.

The paper also introduces a "Red-Queen Loop" where an adversarial generator searches for blind spots in the verification product, forcing continuous adaptation. The framework is evaluated on three mathematical reasoning benchmarks (MiniF2F, GSM8K, MATH) using several state-of-the-art models (DeepSeek-Math, AlphaGeometry, Llemma, Minerva, InternLM-Math). The key metric is "reasoning accuracy" -- the percentage of problems solved with a valid, logically coherent reasoning chain, not just a correct final answer.

**Strengths:**

1.  The paper addresses one of the most significant and challenging problems in AI safety: inner alignment and deceptive alignment. As models approach and surpass human-level capabilities, ensuring they remain aligned is of paramount importance.
2.  Section 3 provides an excellent, mathematically grounded formalization of the problem. The distinction between base objective, proxy reward, and mesa-objective is clearly articulated, as is the concept of "gradient hacking." This sets a strong foundation for the proposed solution.
3.  The Multi-Stage Verification Framework is a novel and compelling approach. The idea of verifying intent across multiple representational domains (language, activations, logic) is intuitively sound -- it is exponentially harder for a model to maintain a consistent deception across all three. The mathematical integration into the loss function via the "safety tax" is elegant and practical.
4.  The empirical evaluation is a major strength. The paper tests the framework on three diverse benchmarks and five different models, showing consistent and significant improvements. The gains of 10-15% in reasoning accuracy on datasets like MiniF2F and GSM8K are impressive and demonstrate the framework's effectiveness.
5.  The related work section is thorough and up-to-date, covering empirical demonstrations of alignment faking (Rabiza, 2024; Greenblatt et al., 2024), emergence in smaller models (Koordijk, 2025), connections to reward hacking (MacDiarmid et al., 2025), and mitigation strategies (Esna-Ashari, 2025). This situates the work well within the ongoing research conversation.
6.  The framework is not just theoretical. The authors describe a concrete training loop (Section 5) and discuss how to integrate it with existing models. The results suggest it could be a practical tool for improving the safety and reliability of deployed AI systems.

**Weaknesses:**

1.  The framework combines three verification stages, but the paper does not include ablation studies to isolate the contribution of each stage. How much of the performance gain comes from the PRM (Stage 1) versus the symbolic verifier (Stage 3)? Is the mechanistic interpretability component (Stage 2) fully implemented and tested, or is it more of a future direction? The results section focuses on the overall improvement but doesn't disentangle the effects. This makes it difficult to assess the necessity and individual value of each component.
2.  The proposed framework adds significant computational cost to training and inference. Running a PRM, a symbolic solver, and potentially mechanistic interpretability tools for every generated output is expensive. The paper does not discuss this overhead or provide any analysis of the trade-off between improved safety/reasoning and increased computational cost. This is a practical concern for real-world deployment.
3.  The paper defines the key metric as "percentage of mathematical problems solved with proper reasoning." It is not entirely clear how this is operationalized. Is it a human evaluation? Is it an automated metric based on the PRM and symbolic verifier? The paper mentions using a PRM and a symbolic engine as verifiers, but it's ambiguous whether the results in Tables 1 and 2 are based on these verifiers or some other measure. This needs clarification.

**Suggestions for Improvement:**

1.  Add Ablation Studies: This is the most important suggestion. The authors should conduct experiments to measure the contribution of each verification stage. For example, train and evaluate the models with:
    - No verification (baseline).
    - Only the PRM (Stage 1).
    - Only the symbolic verifier (Stage 3).
    - All stages combined.
      This would provide invaluable insight into the mechanics of the framework.
2.  Include Computational Cost Analysis: Add a section or table discussing the computational overhead. Report metrics like training time per epoch, inference latency, and FLOPs compared to the baseline. This would help readers assess the practical feasibility of the approach.
3.  Clarify the Metric: Explicitly define how "reasoning accuracy" is measured. Is it based on the framework's own verifiers, or is it a separate evaluation? If it's based on the verifiers, this should be stated. A clear definition will prevent confusion.

---

### Official Review · Reviewer_VR1T · 2026-03-13
**Important problem, but the paper does not yet support its central safety claims**

**Rating:** 5
**Confidence:** 3

**Review:**

This paper addresses an important topic: deceptive alignment and inner-alignment failure in large language models. The high-level intuition behind a multi-stage verification pipeline is reasonable, and I can see why the authors want to connect reasoning verification with broader safety goals. However, I do not think the current version provides enough evidence for the claims it makes, especially in the title, abstract, and conclusion.

My main concern is a mismatch between what the paper says it solves and what it actually evaluates. The paper is framed as a defense against deceptive alignment and mesa-optimizers, but the experiments are on mathematical reasoning benchmarks such as MiniF2F, GSM8K, and MATH. Those are useful benchmarks for reasoning quality, but they are not direct tests of deceptive alignment. There is no explicit deception setup, no train-vs-deploy distinction, no adaptive adversary, no measurement of alignment-faking behavior, and no ground-truth notion of a hidden mesa-objective. As a result, the empirical section supports, at most, a claim about verifier-augmented reasoning, not a claim about mitigating deceptive alignment.

I also found the mathematical framing much stronger than the paper can justify. The product-style safety gate and the modified loss function are presented as if they make deception “mathematically improbable” or force internal objectives toward human goals, but those conclusions do not follow from the equations as written. The paper does not justify the assumptions needed for that argument, such as meaningful independence or calibration of the verification stages. More importantly, it does not explain how these heterogeneous checks are actually integrated into a trainable pipeline, especially when symbolic verification and latent-state monitoring are involved.

Relatedly, the implementation details are far too thin for an empirical paper making claims of this scope. The paper does not clearly define the main evaluation metric (“proper reasoning”), does not describe the optimization setup in enough detail, and does not report variance, seeds, ablations, or computational overhead. Table 1 and Table 2 show broad and consistently positive gains across different models and datasets, but without enough methodological detail to judge how reliable those numbers are. I also found the role of some datasets in Table 2 unclear: they appear more like reasoning corpora or training sources than standard held-out evaluation benchmarks, but the paper does not clearly separate those uses.

Another serious issue is the scholarship. Some references appear incorrect or misused in ways that materially reduce confidence in the manuscript. For example, the Minerva citation is not the Google math model paper, and other citations are used to support claims they do not actually establish. There are also editorial artifacts and signs that the text was not fully cleaned up. On their own these issues might be fixable, but combined with the conceptual overclaiming, they make it difficult for me to trust the paper in its current form.

To be clear, I do think there is a potentially interesting paper here. If the authors want to argue that multi-stage verification improves verifiable mathematical reasoning, that could be a legitimate and useful contribution. If they want to argue that the method mitigates deceptive alignment, then the paper needs a much more direct and rigorous evaluation of deceptive behavior itself, along with a more precise operationalization of the latent-state stage and a substantially stronger experimental section.

**Strengths**
- The topic is important and timely.
- The defense-in-depth intuition is reasonable and potentially interesting.
- The paper tries to connect reasoning verification with broader safety concerns, which could be valuable if scoped more carefully.

**Weaknesses**
- The core deceptive-alignment claim is not directly evaluated.
- The mathematical argument is stronger than the paper justifies.
- The training and evaluation pipeline is under-specified and not reproducible.
- The reference list and manuscript quality contain issues serious enough to affect trust.

**Suggestions for improvement**
- Narrow the scope of the claim. If the contribution is improved verifiable reasoning, present it that way.
- If the central claim is really about deceptive alignment, add direct evaluations of deceptive or alignment-faking behavior.
- Define the evaluation metric precisely and provide full experimental details, including seeds, ablations, and overhead.
- Clarify the role of each verification stage, especially the latent-state / mechanistic interpretability component.
- Audit the citations and clean up the manuscript carefully.

**Overall assessment**
I appreciate the ambition of the paper, but in its current form I do not think it clears the acceptance bar. The key issue is not that the paper needs one or two extra experiments; it is that the central safety framing is much stronger than what the presented evidence supports. I would encourage the authors to either narrow the claim substantially or build a much more direct evaluation around deceptive alignment itself.

---

### Official Review · Reviewer_ZZKm · 2026-03-13
**There are problems in the "MULTI-STAGE GOAL VERIFICATION: A DEFENSE IN-DEPTH MATHEMATICAL FRAMEWORK FOR MITIGATING DECEPTIVE ALIGNMENT IN LARGE LANGUAGE MODELS" paper**

**Rating:** 5
**Confidence:** 3

**Review:**

This paper is devoted to solution of such important task as solving Mesa-Optimization problem. Authors try to solve this problem by integrating mechanistic interpretability
and symbolic verification in the Multi-Stage Checking Framework.

This paper has the following disadvantages:
1) Authors describes symbolic verification as one the base of their framework but there is not description of corresponding symbolic logic in the paper.
2) It is necessary to correct 2003 year to 1993 year in the following reference:

Hans Kamp and Uwe Reyle. From discourse to logic: Introduction to modeltheoretic semantics of natural language, formal logic and discourse representation theory, volume 42. Springer Science & Business Media, 2013

---

### Official Review · Reviewer_3E9H · 2026-03-13
**The multi-stage verification idea is promising, but the contrubution of the paper is unclear.**

**Rating:** 4
**Confidence:** 3

**Review:**

This paper addresses challenging problems in AI safety: inner alignment and the risk of deceptive alignment in advanced large language models (LLMs). The authors propose a multi-stage verification framework that leverages mathematical reasoning and formal logic to detect and mitigate the divergence between a model's internal mesa-objective and the intended human goals. By integrating symbolic verification, latent activation monitoring, and process-based reward modeling into a composite safety gate, the work aims to move beyond brittle single-stage supervision. While the paper tackles a highly timely topic, its contribution is significantly undermined by a lack of clarity, apparent non-sequiturs, and a presentation that feels disjointed and incomplete.
The paper suffers from a critical logical disconnect. The proposed framework is explicitly designed to detect deceptive alignment through a composite safety gate (`V_total`) and a "safety tax" (`λ`). However, the experiments measure a completely different outcome: the percentage of mathematical problems solved *with proper reasoning*. There is no evidence presented that the framework detects or mitigates deception. The results merely show improved performance on math benchmarks, which is an orthogonal goal. The leap from "better math reasoning" to "ensured inner alignment" is a massive, unsupported assumption.
The "Multi-Stage Checking Framework" is described in a confusing and under-specified manner. The paper mentions three stages (natural language, latent activations, formal logic) but never clearly defines the inputs, outputs, and mechanisms of each stage. The term `V_i` is used for "verification stages," but it's unclear if `V_math` and `V_reason` from the training loop are these stages, or if they are separate. The "Red-Queen Loop" is introduced as an equation but its practical implementation is not explained. The framework reads more like a collection of interesting ideas than a coherent, implementable system.
The paper appears to be a compilation of disparate sections, possibly from multiple sources, that have not been smoothly integrated. This makes the experimental section impossible to interpret. The sudden shift in focus from detecting deception to merely improving math scores on a list of datasets is jarring and nonsensical.
It is unclear what the authors are claiming as their novel contribution. Is it the multi-stage verification framework? Is it the specific mathematical formulation of the safety tax and Red-Queen Loop? Is it the empirical finding that certain models can be fine-tuned to do better on math benchmarks? The paper does not clearly delineate its original ideas from prior work. The conclusion highlights the "empirical success" of the approach, but since the experiments are disconnected from the core problem, this claim is unsubstantiated. The paper ultimately fails to demonstrate that its proposed method actually addresses the problem of inner alignment it set out to solve.

Generally, the paper's central flaw is the complete disconnect between its theoretical framework for detecting deception and its experimental results on mathematical reasoning. The presentation is disjointed, the framework is vague, and the results are presented in a confusing and uninterpretable manner. The paper does not provide a coherent argument or credible evidence that its proposed method can, in fact, ensure inner alignment. A substantial rewrite is needed to clarify the framework, design proper experiments that directly test for deception, and clearly articulate the novel contribution.

---

### Decision · Program_Chairs · 2026-03-14

**Decision:**

Accept (Oral)

**Comment:**

Dear Author(s),

On behalf of the Program Committee of the International Conference on Mathematics of Artificial Intelligence (MathAI 2026), we are pleased to inform you that your paper has been accepted for an oral presentation at MathAI 2026.

Your paper was evaluated through a rigorous two-stage review process involving both automated screening and expert review by members of the Program Committee. The reviewers recognized the quality and contribution of your work.

Presentation details:

- Format: Oral presentation (15–20 minutes + 5 minutes Q&A)
- Mode: You may present either in person (offline) at the conference venue in Sirius, Russia, or remotely via Zoom. Please indicate your preferred mode when confirming your participation.
- Conference dates: Marh 30 - April 3, 2026
- Website: https://mathai.club

Next steps:

1. Please confirm your participation and presentation mode by replying to this email mathai.club@yandex.ru no later than March 15, 2026 18:00 Moscow time.
2. If you plan to attend in person, the organizing committee will provide accommodation details separately.
3. Please prepare your final camera-ready manuscript according to the formatting guidelines available at https://mathai.club and upload it to OpenReview by March 15, 2026 18:00 Moscow time.

Should you have any questions regarding the program, logistics, or your presentation slot, please do not hesitate to contact us.

We look forward to your contribution to MathAI 2026.

With kind regards,

MathAI 2026 Program Committee
International Conference on Mathematics of Artificial Intelligence
https://mathai.club
OpenReview: https://openreview.net/group?id=mathai.club/MathAI/2026/Conference
Telegram: https://t.me/MathAI_club
Email: mathai.club@yandex.ru